# A longitudinal observational study of home-based conversations for detecting early dementia: protocol for the CUBOId TV task

Daniel Paul Kumpik ,[1] Raul Santos-Rodriguez,[1] James Selwood,[2,3]
Elizabeth Coulthard,[2,4] Niall Twomey,[5] Ian Craddock,[5] Yoav Ben-Shlomo[3]

¹Department of Engineering Mathematics, University of Bristol, Bristol, UK
²Bristol Medical School, University of Bristol, Bristol, UK
³Department of Population Health Sciences, University of Bristol, Bristol, UK
⁴Department of Translational Health Sciences, University of Bristol, Bristol, UK
⁵Department of Electrical and Electronic Engineering, University of Bristol, Bristol, UK

**Correspondence to**
Daniel Paul Kumpik;
dan.kumpik@bristol.ac.uk

## ABSTRACT

**Introduction** Limitations in effective dementia therapies mean that early diagnosis and monitoring are critical for disease management, but current clinical tools are impractical and/or unreliable, and disregard short-term symptom variability. Behavioural biomarkers of cognitive decline, such as speech, sleep and activity patterns, can manifest prodromal pathological changes. They can be continuously measured at home with smart sensing technologies, and permit leveraging of interpersonal interactions for optimising diagnostic and prognostic performance. Here we describe the ContinUous behavioural Biomarkers Of cognitive Impairment (CUBOId) study, which explores the feasibility of multimodal data fusion for in-home monitoring of mild cognitive impairment (MCI) and early Alzheimer's disease (AD). The report focuses on a subset of CUBOId participants who perform a novel speech task, the 'TV task', designed to track changes in ecologically valid conversations with disease progression.

**Methods and analysis** CUBOId is a longitudinal observational study. Participants have diagnoses of MCI or AD, and controls are their live-in partners with no such diagnosis. Multimodal activity data were passively acquired from wearables and in-home fixed sensors over timespans of 8–25 months. At two time points participants completed the TV task over 5 days by recording audio of their conversations as they watched a favourite TV programme, with further testing to be completed after removal of the sensor installations. Behavioural testing is supported by neuropsychological assessment for deriving ground truths on cognitive status. Deep learning will be used to generate fused multimodal activity-speech embeddings for optimisation of diagnostic and predictive performance from speech alone.

**Ethics and dissemination** CUBOId was approved by an NHS Research Ethics Committee (Wales REC; ref: 18/WA/0158) and is sponsored by University of Bristol. It is supported by the National Institute for Health Research Clinical Research Network West of England. Results will be reported at conferences and in peer-reviewed scientific journals.

## STRENGTHS AND LIMITATIONS OF THIS STUDY

⇒ We characterise diagnostic trajectories across a broad range of behavioural domains relevant to cognitive decline 'in the wild'.
⇒ Patients' live-in partners are contextually matched controls, accounting for differences across participants' home environments and lifestyles.
⇒ Neuropsychological measurements form ground truths for behavioural biomarkers of cognitive decline.
⇒ Limited sample size and restricted demographics may confer limited generalisability of findings to other populations.
⇒ Behavioural and neuropsychological testing timelines disrupted and desynchronised due to COVID-19.

## INTRODUCTION

Alzheimer's disease (AD) is one of the most common neurological diseases, affecting an estimated 42 million people worldwide.[1] AD causes a progressive loss of memory, executive, visuospatial and linguistic functions, which disrupts patients' ability to engage in functional activities of daily living (ADL).[2] Therapeutic options are limited, and clinical interventions are likely more effective in very early or prodromal stages of AD, highlighting the value of early detection.[3] However, abnormal brain pathology can manifest decades before cognitive symptoms[4] and by the time of diagnosis 50% of patients are beyond the early stages.[5]

Mild cognitive impairment (MCI), often a prodromal phase of AD, is characterised by forgetfulness, mood changes, and navigation, decision-making and word-finding problems, which are not sufficiently severe to interfere with ADL.[6] Up to 50% of individuals diagnosed with MCI develop AD within 5 years,[7] but some MCI cases remain stable and some recover over time,[8] complicating prediction of an individual patient's disease progression. Clinical investigations such as lumbar puncture and brain imaging are expensive

and impractical for large-scale deployment,[9] [10] so clinicians have traditionally relied on cognitive screening tests.[11] However, these tests may be limited in validity, specificity, inter-examiner consistency, disregard short-term symptom variability and usually require stressful visits to clinic, which can bias results.[12] [13] They therefore provide a contextually sparse and relatively insensitive measure of subtle early signs of cognitive decline.[4] [13] While blood biomarkers are close to clinical implementation and could revolutionise detection of molecular AD pathology,[14] these are positive early in the course of the illness, so do not mirror clinical decline, and can yield false positives.

These limitations have driven recent interest in passive technologies for longitudinal monitoring of behavioural dementia biomarkers in naturalistic settings.[15–18] Preclinical disruptions in activities such as speaking,[19] environmental navigation,[20] sleep[21] and walking[22] can betray distal biomarkers such as wandering, circadian disturbances, apathy and agitation,[23] which may be useful both in diagnosing MCI and predicting which patients with MCI may go on to develop AD. These ecologically valid, everyday behaviours can be measured continuously and unobtrusively in the home, using networked fixed sensors, or portable sensors in wearables, smartphones and tablets.[16] Fusion across sensing modalities permits fine-grained contextual analysis of how engagement in different ADL covaries over days, weeks and months, and automated machine learning (ML) techniques can leverage intraindividual variability reflective of early cognitive changes.[15] [24] These technologies have potential to help clinicians build personalised and holistic digital phenotypes for prediction of neurodegeneration.[17] [25] However, they remain insufficiently mature for full-scale clinical deployment.[15]

Speech is a particularly interesting dementia biomarker because linguistic competence while cognitively healthy is a known predictor for developing AD.[26] [27] AD disrupts morphological, lexical, semantic, syntactic and structural features of language,[28] and can even be detected in acoustic, temporal and other paralinguistic features.[29–31] Speech is simple to acquire using modern smart devices and therefore is an attractive target for longitudinal in-home monitoring of cognitive decline.[32] [33] Advances in automated speech recognition, natural language processing (NLP) and other ML technologies have contributed to promising results in classifying AD from recorded speech,[34] and recent work has also shown that linguistic and paralinguistic features can help detect MCI.[35] However, classification remains more challenging for MCI than AD, probably because MCI patients' cognitive profiles are heterogenous and not as far removed from those of normal ageing.[29] [36]

Studies leveraging speech to diagnose dementia often train ML models on databases of structured speech gleaned using narrative tasks such as picture description. These provide a rich impression of semantic content, idea density, syntactic complexity, speech fluency and paralinguistic parameters, and discourse fluency,[37] which could be important features for detecting prodromal AD. However, other than occasional interventions from a clinical investigator (which are often discarded (eg,[38])), narrative monologues are constrained by the stimulus material and impoverished in verbal interactions found in natural conversations. Spontaneous conversations require executive attentional, metacognitive and linguistic dexterities not demanded by structured speech tasks that may be useful for detecting MCI due to AD or predicting cognitive decline.[37] [39] Moreover, picture description tasks mostly preclude dyadic signs of cognitive decline, such as prompting in response to word-finding problems.[40] Indeed, the importance of dyadic features for dementia diagnosis is emphasised by the recent finding that it is possible to classify AD from picture description tasks with an accuracy of 78% from the speech of the interviewer alone.[41] While there is acknowledgement that leveraging ecologically valid, naturalistic conversations for diagnosing dementia has merit,[42] [43] speech is rarely collected via passive, *unprompted* observation of patients and their partners 'in the wild', eliciting recent calls for investigation of ecologically acquired, conversational speech relevant to ADL that could confer better diagnostic generalisability.[44]

The ContinUous behavioural Biomarkers Of cognitive Impairment (CUBOId) study at the University of Bristol (UoB) is part of the EPSRC Sensor Platform for HEalthcare in a Residential Environment (SPHERE) Interdisciplinary Research Collaboration.[45] CUBOId aims to advance the state of the art in multimodal in-home sensing for novel dementia-related behavioural biomarkers[46] such as sleep disturbances, partner shadowing, wandering and disrupted conversational speech. We will develop novel computational algorithms to infer biomarkers of cognitive decline as measured using sensor networks in MCI and AD patients' own homes, permitting autonomous, continuous, long-term behavioural sensing. Where possible these will be validated against neuroimaging (amyloid positron emission tomography (PET)) data and neuropsychological progression over time. The patient's live-in companions are also participants, to act as environmentally matched controls and shed light on interactional biomarkers of cognitive decline. Because the companion is familiar to the patient, we hope to record behavioural interactions that have high ecological validity.

Here we report on the data acquisition and analysis protocol for CUBOId's novel conversational speech task, the 'TV task'. We give an overview of CUBOId's in-home sensing methods and outline how concurrent sensor readings from environmental sensors, cameras and wearable accelerometers can be combined, and further fused with conversational speech features, to improve diagnosis from speech alone. Our objectives are as follows:

1. To develop ML models capable of leveraging home-acquired, natural speech for automated prediction, diagnosis and longitudinal tracking of cognitive decline.

For this we will establish ground truths on early speech biomarkers of MCI/AD from neurological and neuro-psychological tests.

2. To investigate how temporal variability in linguistic proficiency is reflected in clinically relevant variability in ADL behaviours. For example, we hypothesise that periods of agitation, wandering and sleep disturbances will precede disturbed speech as revealed by acoustic and prosodic changes, hesitations, mnemonic search markers and endogenous (eg, rapid speech) and ex-ogenous (eg, calming conversational interactions) lin-guistic features.

3. To use insights from objective 2 and data fusion tech-niques to leverage our multi-modal data streams by deriving latent relational embeddings between speech and multiple sensor readouts for optimisation of diag-nostic and predictive performance from speech alone.

While CUBOId's sensing regime is intended to be comprehensive, we aim to define the minimum dataset required for disease detection and monitoring, to enable more cost-efficient future use with simpler data capture methods. A further outcome will be pilot data on effect sizes sufficient to generate a meaningful power calcula-tion to guide future studies.

## METHODS
This is a longitudinal observational 'proof of concept' study to examine speech and other behavioural biomarkers among cases with MCI or early AD and their partners, who act as controls. Speech is repeatedly measured over time in parallel with repeat neuropsychology and contin-uous multimodal home sensing. This permits investiga-tion of whether speech-based diagnosis and monitoring of early dementia can be optimised using simultaneously recorded behavioural data. CUBOId started on 22 March 2019 and runs until 31 December 2022.

### Patient and public involvement
Prior to CUBOId deployments, SPHERE ran co-design workshops and publicly trialled its consent processes. On average, 86% properly understood the risks and 92% consented to take part. Patients were involved in pilot experiments for CUBOId's behavioural tasks. A patient and public involvement (PPI) advisory team gave recom-mendations on recruitment methods, materials provided to participants and other practical issues. We plan to involve the PPI team with dissemination of materials to participants, the public and healthcare professionals.

### Participants and eligibility criteria
Although data collection is ongoing, recruitment has now ended. Participants were recruited via the cognitive neurology service at the North Bristol NHS Trust Brain Centre, the Bristol Dementia Wellbeing Service and the Research Institute for the Care of Older People in Bath. Informed consent was obtained by a clinical fellow (JS) at the North Bristol Trust Brain Centre.

Participant pairs, each consisting of an outpatient with a previous diagnosis of either MCI or AD (confirmed by JS and EC) and their partner or live-in carer with no such diagnosis, were eligible. Participants with MCI were offered MRI and amyloid PET brain scans on a voluntary basis to discern the likely aetiology of their cognitive complaint (eg, AD, vascular disease, subjective cognitive disorder). Where available, these were complemented with routine cerebrospinal fluid measurements of amyloid-β 42 and total tau expressed as total tau/amyloid-β 42 ratio, using a validated cut-off used in clinical practice above which an AD diagnosis is thought likely.

Participants with severe comorbidities were not consid-ered. Participants were selected according to the following criteria: (a) full capacity to give informed consent; (b) at least 50 years old; (c) competent English speaker; (d) normal or corrected-to-normal vision; (e) consent to have SPHERE installed in their home.

A conventional sample size calculation was not performed as CUBOId is a precursor to future larger scale evaluations if the results look hopeful. From the larger CUBOId participant pool nine sets of participants are participating in the TV task, of which eight are patient-companion pairs with one MCI or AD patient (4/8 female). All eight participant pairs have one male and one female participant. An additional male MCI patient lives alone, but nevertheless verbally describes the TV show during each recording session, permitting insight into how TV task performance differs under monadic and dyadic conditions. The mean age (±SD) of MCI and AD patients at study entry was 71.4±6.9 years, and the mean age of companions was 64±14 years (this informa-tion was not available for one companion). Companions are usually the life partners of the patients (one case is an adult child who lives with their parent). 6/9 patients had prior MCI diagnoses and 3/9 had prior AD diagnoses.

Some participants have hearing problems, a known risk factor for cognitive decline. We will therefore issue participants with a custom hearing questionnaire derived from the Speech, Spatial and Qualities of Hearing ques-tionnaire (short form)[47] and the Glasgow Hearing Aid Benefit Profile[48] (see online supplemental material S1).

### The novel paradigm of the TV task
This task was inspired by the UK TV show 'Gogglebox'. Participants record their conversations as they watch a TV programme of their choice such as the news, soap operas or other regular shows. They do so for a minimum of 30 min, at roughly the same time of day, over five consecutive days. It is emphasised that these are ideal-ised guidelines and that participants should prioritise daily activities and unforeseen events over the TV task recording schedule. Participants are therefore asked, at minimum, to provide five half-hour recordings regardless of spacing between recordings. To catch discrepancies in TV shows or timings of recordings across days, partici-pants recite the date and time and the name of the show at the beginning of each recording.

Participants place the tablet between their normal sitting positions for watching TV, then activate the audio recording app and converse naturally while using the TV programme as a stimulus for conversation. This guideline is to emphasise our desire to acquire naturalistic conversation, but also to maximise the amount of speech we record. The nature of the conversation is largely determined by the TV show, with for example, news, geographical culture shows, gameshows and soap operas stimulating conversations that are, respectively: topical, about places and holidays, participatory and collaborative, or commentaries on the story and characters. Participants are asked to leave the app recording during adverts and if possible, until the end of the show. Once participants acquire five half-hour recordings, they return the tablet to the CUBOId team via courier.

Speech recordings are made using a Samsung Galaxy SM-T580 tablet and a preinstalled app (https://github.com/Dimowner/AudioRecorder), chosen because it is free of advertisements and easy to use, while providing a good range of recording options. Audio files are recorded at a sample rate of 44.1 kHz in mono in WAV format, apart from one pilot data collection block, which was recorded in MP3 format.

Tablets were provided by the research team during home visits prior to COVID-19's arrival, but are now couriered to participants. Couriered tablets, chargers and parcel tape for return packaging are disinfected with anti-viral spray, and a minimum 3-day period is enforced after packing before participants open the package. Then, at a pre-agreed time, participants are contacted by the CUBOId team and given training on the TV task either over the phone or securely using Zoom video conferencing software.[49]

Participants are shown how to use the Audio Recorder app and how to delete a recording if necessary for privacy reasons. After this, companions and patients provide controlled speech recordings by reading aloud ten sentences from the Harvard list,[50] a set of phonetically balanced English sentences used to assess speech intelligibility in telecommunications systems. These control recordings provide a template for training deep neural networks to diarise participants' speech.

Participants then perform the TV task at their convenience. Detailed instructions describing the recording procedure are included as a PDF file on the device home screen, and telephone support is available in case of unforeseen difficulties.

### Impact of COVID-19 on TV task data collection timeline

COVID-19, and technician/researcher availability, have caused severe disruptions to CUBOId by desynchronising the behavioural tasks. We originally intended to collect behavioural data across 3–4 testing blocks per participant pair, held at roughly 0, 6 and 12 months after SPHERE installation, with the possibility of a fourth block at 18 months if possible. We intended to visit participants' homes to train them on the behavioural tasks, but we suspended SPHERE deployments in March 2020 and introduced remote testing protocols to mitigate risk to participants. One participant pair first performed the TV task prior to the pandemic and had a large gap (17 months) between the first and second TV task administrations. The remaining participant sets completed the first TV task block in January to March 2021, after which one set withdrew from the TV task. All remaining participant sets completed their second block in July to September 2021, six of which still concurrently had active SPHERE deployments installed. A third testing block will be completed before 31 December 2022, without sensors present in participants' homes. To date, nine participant sets have completed TV task testing block 1, eight have completed block 2 and block 3 is currently underway.

### Neuropsychological and cognitive tasks

Participants completed baseline neuropsychological testing (administered by JS: see online supplemental material S2) before SPHERE installation. We intended to conduct repeat assessments at 6 and 12 months follow-up, but COVID-19 restrictions delayed these until later in the study. The test battery was adapted from the recommendations of the European Prevention of Alzheimer's Dementia study.[51] Participants also completed the Alzheimer's Disease Assessment Scale-Cognitive subscale (ADAS-Cog), considered the gold standard for dementia research trials.[52]

Participants' psychological well-being and sleep were assessed using the 6-Item State-Trait Anxiety Inventory,[53] the Geriatric Depression Scale (Short Version),[54] the Epworth Sleepiness Scale[55] and the Pittsburgh Sleep Quality Index.[56] After COVID-19's arrival, neuropsychological and cognitive assessments were conducted using the online video-conferencing platform Attend Anywhere (https://www.attendanywhere.com).

To measure memory performance participants complete a memory task at home using the Mezurio smartphone app[57] after each TV task testing block. Participants are encouraged to play Mezurio at roughly the same time each day, for about 10 min per day, for 30 days.

### Multimodal sensing with SPHERE

SPHERE sensor modalities are categorised as wearable, environmental or silhouette. Custom wrist-worn wearables record acceleration indicative of bodily activity, posture and movement, and room-level location via triangulation of received signal strength measurements at Bluetooth access points throughout the home. The environmental modality consists of room sensors for occupancy, temperature, humidity, water and electricity usage. Silhouette cameras yield whole-body pose as well as fine-grained location information while obfuscating identity to preserve privacy.

### DATA ANALYSIS

We will use Python and associated libraries for all analyses unless otherwise stated. Features derived from SPHERE

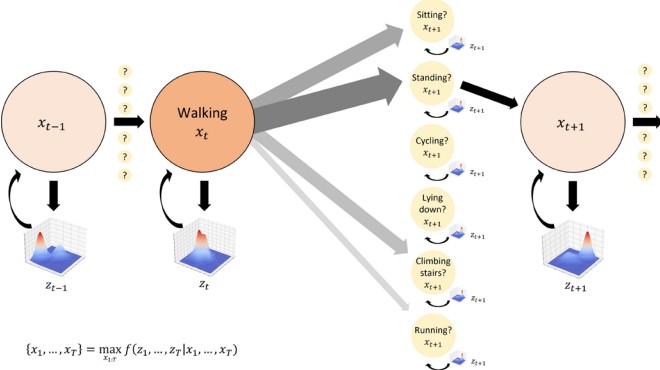

$$\{x_1, ..., x_T\} = \max_{x_{1:T}} f(z_1, ..., z_T | x_1, ..., x_T)$$

**Figure 1** Hidden Markov Models for activity detection. Activity transitions can be modelled as state changes in a Markov chain, with hidden activity states $x_{ti}$ at time $t$ represented by observed sensor (eg, wearable accelerometer) inputs $z_{ti}$. Activities are assumed to transition sequentially with fixed probabilities obtained empirically (shown by the thickness and shading of the grey arrows): in this example, walking generally transitions to standing more often than to sitting, and never directly to lying down or cycling because a standing transition is required first. Maximum likelihood estimation over the entire sequence of sensor inputs is used to obtain an optimal activity transition sequence.

sensors and speech recordings will be selected by examining their contributions to the variance in each dataset, their correlations with clinical measures such as ADAS-Cog and from insights into how ADL and speech covary longitudinally with disease progression. After standardisation of data by the within-population standard deviation, effect sizes for power calculations in future studies will be estimated from differences between (a) controls (partners) and patients; (b) houses with MCI and AD patients; and (c) baseline performance and follow-up testing or analysis (for progression over time).[58]

### Location and activity detection from SPHERE sensor data
Dedicated preprocessing pipelines will be applied for different sensor types, but fusion across sensors will provide a spatially and temporally contextualised overview of ADL behaviours and anomalies. We will explore which sensors are best combined for different activities and at what level of integration, and also the optimal stages at which to fuse activity with location data, ranging from early (naïve multimodal feature set integration) to late (hierarchical integration of posterior probabilities from deep learning models trained on individual feature sets) fusion strategies.[24 59]

Activity detection and in-home localisation will leverage the sequential nature of our data by recognising that certain behavioural transitions follow others with characteristic probabilities (figure 1). For example, people never transition directly from sitting to running, but often from sitting to standing and then running. Similarly, changes in participant location can only proceed across adjacent positions in physical space. This sequential dependence of states can be modelled using Viterbi decoding, which tracks a maximum-likelihood path through a series

of noisy, probabilistic transitions in a Hidden Markov Model (HMM).[59 60] We will compare HMM activity classification and localisation performance to deep learning architectures capable of leveraging spatial and temporal relationships in the multimodal feature space such as convolutional neural networks and long short-term memory networks.[24] We will compare these with pipelines exploiting both activity and location data together for further disambiguation of behaviour.

At the highest analysis level, we will derive sensor feature combinations suggestive of periods of 'abnormal' dementia-related behaviours. For example, wandering could be inferred from a joint comparison of the wearable's activity levels and location complexity.[46]

Behaviours we have identified as likely to be indicative of cognitive decline include[2 46]:
► Wandering and shadowing of the companion.
► Sleep disturbances.
► Agitation.
► Apathy, sedentarisation.

We will analyse these longitudinally to identify short periods of unusual behaviour and longer-term changes that may be indicative of disease progression and anomalous events in the recent past or future that could be related to such changes, such as poor sleep during nights before an episode of agitation. To leverage these behaviours alongside our conversational speech data, we will use representation learning to generate novel latent activity/speech embeddings for dementia detection based on speech alone. We will project these novel features onto unseen speech samples after exploring the best subset of SPHERE sensors to combine with our paralinguistic and linguistic models, and also whether diagnostic performance is better at this low level of SPHERE sensor integration compared with, for example, combining speech with actograms derived from preintegrated sensor data. Fusion techniques will include multiview and transfer learning.

### Speech preprocessing
We will build deep learning pipelines comprising paralinguistic and NLP domains to detect MCI, and predict the probability of progression to AD, from conversational speech, after experimenting with low and high-level fusion strategies for paralinguistic and linguistic features. The speech recordings suffer from quality issues that will require a dedicated preprocessing and speaker diarisation pipeline. First, the samples contain two participants who speak infrequently (and sometimes over each other) against a variable TV background. The TV produces structured sounds including music, speech and sound effects with unpredictable spectrotemporal statistics that often resemble the participants' voices (figure 2). Our single-channel recording setup is challenging for standard source separation techniques, but because we have acquired controlled voice recordings from each participant, we will use a semi-supervised learning architecture (such as variational autoencoders) to derive latent

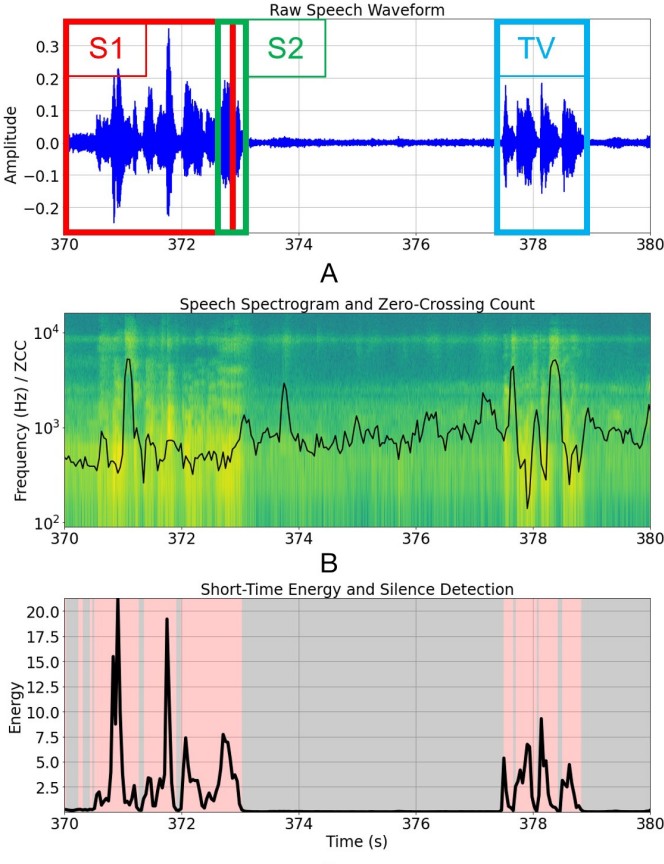

**Figure 2** TV task speech preprocessing for speaker diarisation. (A) Raw speech waveform with overlapping speakers 1 and 2 labelled, as well as the TV. (B): Speech spectrogram with zero-crossing count, a feature useful for detecting voiced and unvoiced speech, overlaid. (C) Silence removal using a naïve energy detector after careful denoising.

features for target separation. PRAAT phonetics software[61] will be used for annotating participants' voices for evaluation purposes, and recordings will be transcribed for NLP modelling according to established protocols.[62]

### Acoustic and paralinguistic features

We will extract non-linguistic speech features at local (ie, fast Fourier transform window duration), semi-local (ie, central tendency and variability statistics of features calculated on time scales of seconds) and global (ie, over an entire audio file) time scales. Guided by feature taxonomies in recent reviews,[44 63] local features will include representations of the speech spectrogram, such as Mel-frequency cepstral coefficients, as well as spectral bandwidth, short-time energy and zero-crossing count. Semi-local features will comprise paralinguistic measures of vocal quality known to reflect neuropathology such as jitter and shimmer, and higher prosodic features such as intonation, stress and rhythm. At the global feature level, we will compute utterance statistics such as articulation rate, time spent talking and mean utterance duration, and temporal mnemonic features relating to hesitations. We will extract interactional and pragmatic features

such as the number of turn changes and overlapping segments,[64] number of repair sequences[65] and inter-turn silence duration.[66]

### Natural language processing

We will analyse transcripts of CUBOId participants' speech using NLP modelling and deep learning. Python's Natural Language Toolkit and similar open resources will be used to conduct analyses on cognition-relevant features at the lexical, connected language and semantic levels.[28] At the lexical level, tokenisation and stemming, morphological parsing and part-of-speech tagging will help derive measures of lexical diversity. At the level of connected language, we will explore pragmatics, syntactic complexity and correctness, topic coherence, and sequence predictability measures of disordered thinking. At the semantic level we will characterise idea density, a measure of linguistic conceptual richness highly predictive of cognitive decline in later life, and we will experiment with word embeddings selected from pretrained language models such as BERT,[67] which have found a cutting-edge role in dementia detection studies.

### Ethics and dissemination

A major issue for studies with vulnerable people is to ensure they understand and are willing participants. Therefore JS, a clinician with extensive experience of working with dementia patients, ensured that all participants had capacity to consent at the outset. Given CUBOId's longitudinal nature, it is possible that capacity may be lost over time. Should this occur the partner is asked their opinion as to whether or not it is appropriate to proceed based on the patient's previously stated preferences.

CUBOId is sponsored by UoB and was approved by the Wales Research Ethics Committee (REC) 7 (ref: 18/WA/0158) and the Health Research Authority (IRAS project ID 234027). CUBOId is supported by the National Institute for Health Research (NIHR) Clinical Research Network West of England. Protocol amendments and adverse events are referred to the REC and UoB. UoB has Public Liability and Professional Negligence insurance policies to cover the eventuality of harm to a research participant or University employee arising from design or management of the research. Unforeseen protocol deviations are reported to the Principal Investigator, and serious deviations to the Sponsor, immediately. If CUBOId is ended prematurely, the Principal Investigator will notify the UoB and Wales RECs, including the reasons for termination.

### Data statement

All individuals in the CUBOId team have unrestricted access to the full dataset. Results will be reported at conferences, in peer-reviewed scientific journals and on the UoB website.

Both the anonymised data and study documentation are stored on secure servers within UoB and shared according to UoB procedures and guidelines. Personally

identifiable information is stored separately to research data. Data management procedures follow the stipulations of the General Data Protection Regulation. In line with NIHR guidance, we obtained consent for participants' data to be stored for up to 20 years, and deidentified data will be shared in the data repository by the SPHERE team to bona fide researchers with ethical approval for aligned projects, subject to a legally binding data agreement (check with the SPHERE team about the availability of the data through the repository after the project ends).

Privacy issues relating to identification of participants from CUBOId video recordings are addressed by using cameras that retain only the silhouettes of participants before data transfer to SPHERE servers. Analogous concerns regarding TV task audio recordings are addressed through careful curation and highlighting of sensitive information at annotation time for future obfuscation of, for example, names with aliases. If audio data are released in future the voices may be further obfuscated using neural anonymisation techniques.

## Summary

To our knowledge there is no full-scale deployment of in-home, real-world, longitudinal activity sensing technologies that is complemented by monitoring of naturalistic conversations and neuropsychological assessment of MCI or AD patients. By bringing together these behavioural domains, CUBOId will observe how they covary with disease progression, permitting construction of models that perform better than those trained on a single modality. Here, we present a novel protocol for recording speech from MCI and AD patients and their companions as they converse at home. We intend to develop, test and de-risk a novel speech-based diagnostic asset for the UK Dementias Research Institute community, which may be applied to cohorts in the Dementia Platform UK, and the international community.

**Acknowledgements** We would like to thank Julia Carey, David Bailey and Danielle Hale for their support in administering CUBOId and in collecting data. We would also like to thank our participants for their involvement in CUBOId.

**Contributors** YB-S, EC, JS, RS-R, NT and IC conceived and designed CUBOId. DPK, YB-S, EC, JS, RS-R and NT designed the TV task, and DPK collected the TV task data and drafted the manuscript. All authors reviewed the manuscript prior to submission.

**Funding** This research was funded by an MRC Momentum award (grant MC/PC/16029), the EPSRC SPHERE Interdisciplinary Research Collaboration (grant EP/K031910/1) and the EPSRC Centre for Doctoral Training in Digital Health and Care, University of Bristol (grant EP/S023704/1). RSR is funded by the UKRI Turing AI Fellowship EP/V024817/1. Analysis of behavioural data collected using SPHERE was partially funded by the BRACE charity.

**Competing interests** None declared.

**Patient and public involvement** Patients and/or the public were involved in the design, or conduct, or reporting, or dissemination plans of this research. Refer to the Methods section for further details.

**Patient consent for publication** Not applicable.

**Provenance and peer review** Not commissioned; externally peer reviewed.

**ORCID iD**
Daniel Paul Kumpik http://orcid.org/0000-0002-5458-4131

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
