## [Reviewer comments · BMJ Open]

ARTICLE DETAILS

TITLE (PROVISIONAL)	A longitudinal observational study of home-based conversations for detecting early dementia: Protocol for the CUBOld TV task
AUTHORS	Kumpik, Daniel; Santos-Rodriguez, Raul; Selwood, James; Coulthard, Liz; Twomey, Niall; Craddock, Ian; Ben-Shlomo, Yoav

VERSION 1 – REVIEW

REVIEWER	Alfalahi , Hessa Khalifa University of Science and Technology
REVIEW RETURNED	02-Sep-2022

GENERAL COMMENTS	In this protocol, Kumpik and colleagues present the plan of the COntinuoud Behavioral Biomarkers Of cognitive Impairment (CUBOld) study that aims to leveraging ML techniques for the detection and longitudinal monitoring of MCI and early AD remotely at home environment; with emphasis on speech data. CUBOld is a longitudinal study of which data collection is still ongoing until 31/12/2022. Interestingly, prior to the study, given the importance of co-design and “co-creation” in the domain of digital medicine [1], the authors performed workshops involving study participants in the pilot experiments of the study. The methodology behind the study is fully explained in an excellent and clear way. I only have the following two points: It is not clear what are the multimodal data that will be fused ? Objectives 2 and 3 in the introduction are a bit confusing given that the notion behind the protocol is to mainly leverage speech analysis. Please clarify more especially in the objectives section of the introduction. It is not clear how, in the opinion of the authors, the minimum dataset for the optimization of meaningful power calculation and for efficient monitoring will be determined, as claimed in the introduction as well. [1] Alfalahi, H., Khandoker, A.H., Chowdhury, N. et al. Diagnostic accuracy of keystroke dynamics as digital biomarkers for fine motor decline in neuropsychiatric disorders: a systematic review and meta-analysis. Sci Rep 12, 7690 (2022). https://doi.org/10.1038/s41598-022-11865-7
---

REVIEWER	Liang, Xiaohui University of Massachusetts System
REVIEW RETURNED	07-Oct-2022

GENERAL COMMENTS	The authors may want to add some more explanation of the content of the conversation. I am not aware of “Gogglebox” and have to google it to understand the task. This task gathers rich audio and video data about user from home, but may raise privacy concerns. The authors may want to refer to the following paper. Liang, Xiaohui, et al. "Evaluating voice-assistant commands for dementia detection." Computer Speech & Language 72 (2022): 101297.
--

VERSION 1 – AUTHOR RESPONSE

Reviewer: 1

Dr. Hessa Alfalahi , Khalifa University of Science and Technology

Comments to the Author:

In this protocol, Kumpik and colleagues present the plan of the COntinuoud Behavioral Biomarkers Of cognitive Impairment (CUBOld) study that aims to leveraging ML techniques for the detection and longitudinal monitoring of MCI and early AD remotely at home environment; with emphasis on speech data.

CUBOld is a longitudinal study of which data collection is still ongoing until 31/12/2022.

Interestingly, prior to the study, given the importance of co-design and “co-creation” in the domain of digital medicine [1], the authors performed workshops involving study participants in the pilot experiments of the study.

- We thank Dr. Alfalahi for pointing out this systematic review, which is relevant background for our paper due to its discussion of keystroke monitoring and other modalities as passive diagnostic sensing technologies for detecting dementia in naturalistic environments. We have cited it in the Introduction (p.4, paragraph 2; reference [18]).

The methodology behind the study is fully explained in an excellent and clear way.

- We thank Dr. Alfalahi for these comments

I only have the following two points:

It is not clear what are the multimodal data that will be fused ? Objectives 2 and 3 in the introduction are a bit confusing given that the notion behind the protocol is to mainly leverage speech analysis. Please clarify more especially in the objectives section of the introduction.

- The three analysis stages of the protocol all involve speech analysis, and indeed for Objective 1 we intend to develop baseline diagnostic models based on speech only, but for Objectives 2 and 3 we will also include data from SPHERE to explore how speech and SPHERE data covary (Objective 2), and how best to fuse data from the different modalities (Objective 3). We agree that this could be made clearer. We have made changes to the Objectives section of the Introduction (pages 5-6). To better define the scope of the paper before listing the objectives (page 5), we have added the text:
 - Here we report on the data acquisition and analysis protocol for CUBOld's novel conversational speech task, the “TV task”. We give an overview of CUBOld's in-home sensing methods and outline how concurrent sensor readings from environmental sensors, cameras and wearable accelerometers can be combined, and further fused with conversational speech features, to improve diagnosis from speech alone.

- The text of Objective 2 (page 6) has been edited to clarify/emphasise that obtaining measures of linguistic proficiency is the initial prime focus of the TV Task study, and to remove some potentially confusing text that is repetitive with Objective 3. The text now reads:
 - 2. To investigate how temporal variability in linguistic proficiency is reflected in clinically relevant variability in ADL behaviours. For example, we hypothesise that periods of agitation, wandering and sleep disturbances will precede disturbed speech as revealed by acoustic and prosodic changes, hesitations, mnemonic search markers and endogenous (e.g., rapid speech) and exogenous (e.g., calming conversational interactions) linguistic features.
- The text of Objective 3 (page 6) has been edited to reflect that data fusion will follow naturally from the investigations of covariance in speech and ADL behaviours planned for Objective 2, and to emphasise that data fusion will be between speech and data from multiple SPHERE sensor readouts. The text now reads:
 - To use insights from Objective 2 and data fusion techniques to leverage our multi-modal data streams by deriving latent relational embeddings between speech and multiple sensor readouts for optimisation of diagnostic and predictive performance from speech alone.
- We have also added some text to the final paragraph on page 11, to expand on how speech and SPHERE sensor fusion will be explored for optimising diagnostic performance. This section now reads:
 - We will project these novel features onto unseen speech samples after exploring the best subset of SPHERE sensors to combine with our paralinguistic and linguistic models, and also whether diagnostic performance is better at this low level of SPHERE sensor integration compared with, for example, combining speech with actograms derived from pre-integrated sensor data. Fusion techniques will include multiview and transfer learning.

It is not clear how, in the opinion of the authors, the minimum dataset for the optimization of meaningful power calculation and for efficient monitoring will be determined, as claimed in the introduction as well.

- We thank Dr. Alfalahi for this observation. We have now provided additional information on how this will be done at the beginning of the data analysis section (page 10), and have provided an additional reference:
 - After standardisation of data by the within-population standard deviation, effect sizes for power calculations in future studies will be estimated from differences between a) controls (partners) and patients; b) houses with MCI and AD patients; and c) baseline performance and follow-up testing or analysis (for progression over time) [58].

[1] Alfalahi, H., Khandoker, A.H., Chowdhury, N. et al. Diagnostic accuracy of keystroke dynamics as digital biomarkers for fine motor decline in neuropsychiatric disorders: a systematic review and meta-analysis. *Sci Rep* 12, 7690 (2022). <https://doi.org/10.1038/s41598-022-11865-7>

Reviewer: 2

Xiaohui Liang, University of Massachusetts System

Comments to the Author:

The authors may want to add some more explanation of the content of the conversation. I am not aware of “Gogglebox” and have to google it to understand the task.

- We thank Dr. Liang for this suggestion. We have clarified the nature of the conversations on page 8, by adding the word “UK” to the first sentence of the section entitled “The novel paradigm of the TV task”, to emphasise that Gogglebox is a UK TV show. We have also added the following text to the second paragraph of that section:

The nature of the conversation is largely determined by the TV show, with e.g., news, geographical culture shows, gameshows and soap operas stimulating conversations that are, respectively: topical, about places and holidays, participatory and collaborative, or commentaries on the story and characters.

This task gathers rich audio and video data about user from home, but may raise privacy concerns.

- This is a very valid point. To emphasise that the silhouette cameras are privacy-preserving, we have amended the final sentence of the section “Multimodal sensing with SPHERE” (page 10) to read:
 - Silhouette cameras yield whole-body pose as well as fine-grained location information whilst obfuscating identity to preserve privacy.
- We have also added a paragraph emphasising privacy mitigations to the “Data Statement” on page 14:
 - Privacy issues relating to identification of participants from CUBOld video recordings are addressed by using cameras that retain only the silhouettes of participants before data transfer to SPHERE servers. Analogous concerns regarding TV task audio recordings are addressed through careful curation and highlighting of sensitive information at annotation time for future obfuscation of, e.g., names with aliases. If audio data are released in future the voices may be further obfuscated using neural anonymisation techniques.

The authors may want to refer to the following paper.

Liang, Xiaohui, et al. "Evaluating voice-assistant commands for dementia detection."

Computer Speech & Language 72 (2022): 101297.

- We thank Dr. Liang for this suggestion, this is indeed a very relevant publication and we have cited it in the Introduction on page 4 (reference [33])

Reviewer: 1

Competing interests of Reviewer: I declare that I have no competing interests

Reviewer: 2

Competing interests of Reviewer: This paper describe a protocol of new data collection that potentially implements early detection of dementia. The data is gathered from spontaneous conversation. The users in the conversations comment on the same TV shows, and their performance can be comparable.